# Variance Reduction in Bipartite Experiments through Correlation Clustering

**Jean Pouget-Abadie**
Google Research
New York, NY 10011
`jeanpa@google.com`

**Kevin Aydin**
Google Research
Mountain View, CA 94043
`kaydin@google.com`

**Warren Schudy**
Google Research
New York, NY 10011
`wschudy@google.com`

**Kay Brodersen**
Google
Zürich, Switzerland
`kbrodersen@google.com`

**Vahab Mirrokni**
Google Research
New York, NY 10011
`mirrokni@google.com`

## Abstract

Causal inference in randomized experiments typically assumes that the units of randomization and the units of analysis are one and the same. In some applications, however, these two roles are played by distinct entities linked by a bipartite graph. The key challenge in such bipartite settings is how to avoid interference bias, which would typically arise if we simply randomized the treatment at the level of analysis units. One effective way of minimizing interference bias in standard experiments is through cluster randomization, but this design has not been studied in the bipartite setting where conventional clustering schemes can lead to poorly powered experiments. This paper introduces a novel clustering objective and a corresponding algorithm that partitions a bipartite graph so as to maximize the statistical power of a bipartite experiment on that graph. Whereas previous work relied on balanced partitioning, our formulation suggests the use of a correlation clustering objective. We use a publicly-available graph of Amazon user-item reviews to validate our solution and illustrate how it substantially increases the statistical power in bipartite experiments.

## 1 Introduction

Whether the setting is a medical trial or an A/B test, causal inference in randomized experiments typically assumes that the units of randomization and the units of analysis are one and the same. One possible exception to this rule is in the context of interference, a field of growing interest to statisticians, where experimenters will occasionally consider the indirect effect of a unit's *peers* being treated. In some cases, it is infeasible or undesirable to assign treatment to the experimental units we care about, and we can then draw a distinction between the units that are directly treated and the units whose outcomes we care about, a setting known as a *bipartite experiment*.

Bipartite experiments are critical in various settings, including pricing in online marketplaces (Fradkin et al., 2018), recommender systems (Gilotte et al., 2018), social networks (Bakshy et al., 2014), and display ad auctions (Chawla et al., 2016). For example, consider an online retailer wishing to determine the impact of offering a discount on certain items through a randomized experiment. The retailer faces the choice of randomizing on users, items, or user-item pairs. Randomizing on users or user-item pairs could result in different users seeing different prices on the same item, which is undesirable. If the retailer decides to randomize on items, standard causal theory would suggest using items as the unit of analysis. However, this ignores interference between substitute goods. For

example, experimenters may be misled by the sudden spike in demand for an item randomly selected for a discount, when in fact the treatment (discount) would be neutral if applied to all of that item's substitutes. An effective way of overcoming these issues is to design and analyze the experiment in a bipartite way, with *users* as the units of analysis and *items* as the units of randomization. Another application arises in recommender systems, which create non-trivial interference mechanisms: boosting the recommendation of one item during a randomized experiment can negatively affect the likelihood of other candidates being recommended to a user. These experiments on such systems may benefit from being analyzed in a bipartite way. A third application is given by display ad auctions (Pouget-Abadie et al., 2018), where bidders compete for ad impressions in auctions, leading to complex interference mechanisms that can be elegantly dealt with using a bipartite experiment.

Despite their importance, the existing literature on bipartite experiments is scarce. Zigler and Papadogeorgou (2018) introduce a formal framework for bipartite experiments and suggest a restriction of the potential outcomes space inspired by the literature on causal inference with interference (Hudgens and Halloran, 2008; Toulis and Kao, 2013; Tchetgen and VanderWeele, 2012). We consider a very similar setting to Zigler and Papadogeorgou (2018) by tying the analysis of bipartite randomized experiments to the dose-response literature, alias continuous treatment literature, more commonly found in medical and public health settings (Galagate, 2016). Extending the idea of cluster-randomized designs to the bipartite setting is the core contribution of the present paper.

In a cluster-randomized design, the treatment is assigned to groups (clusters) of units rather than to individual units. They are a popular design choice for their simplicity and ability to minimize estimation bias (Eckles et al., 2017). Because cluster-based designs are easily implemented and straightforward to analyze, the literature has focused on the choice of clustering, relying either on domain-specific knowledge (e.g. schools districts (Basse and Feller, 2018), villages (Shakya et al., 2017)) or graph algorithms (Ugander and Backstrom, 2013; Gui et al., 2015). A popular clustering technique applied to these settings is balanced graph partitioning (Delling et al., 2012; Stanton and Kliot, 2012; Ugander and Backstrom, 2013; Tsourakakis et al., 2014; Aydin et al., 2016). While some of these clustering techniques can be extended to the bipartite setting, we suggest an entirely new clustering objective—and a corresponding algorithm—that leverages the unique nature of bipartite experiments. In particular, we show how to model our clustering objective as a correlation clustering problem. In doing so, we come closest to the optimal design literature (Raudenbush, 1997; Pokhiko et al., 2019), which selects the experimental design that is optimal according to some statistical criterion.

In Section 2, we define the bipartite randomized experiments framework, and suggest an analysis approach based on a linear treatment exposure assumption. In Section 3, we introduce a new clustering objective for running cluster-based bipartite experiments, and show that it is a specific instance of the well-studied correlation clustering problem. Finally, in section 4, using a publicly-available Amazon user-item review dataset, we show that our suggested algorithm improves experimental power significantly over other more straightforward extensions of cluster randomized designs to the bipartite setting.

## 2 Bipartite randomized experiments

### 2.1 Definitions

A bipartite randomized experiment is a randomized experiment in which we distinguish two types of units: *diversion units* and *outcome units*. Treatment and control is assigned at the level of *diversion units*, and outcomes are measured at the level of *outcome units*. In a traditional randomized experiment, diversion units and outcome units are one and the same; in a bipartite randomized experiment, they are distinct. An outcome unit does not receive any treatment directly. Instead, its observed outcome is determined by the assignment of the diversion units to treatment and control. This dependence is often represented by a bipartite graph—hence the name—linking diversion units to outcome units.

Using potential outcomes notation (Rubin, 2005), let $\mathbf{Z} \in \{0,1\}^M$ be the assignment vector of the $M$ diversion units to treatment ($Z_j = 1$) or control ($Z_j = 0$) and $\mathbf{Y} \in \{0,1\}^M \to R^N$ be the potential outcomes of the $N$ outcome units. For each outcome unit $i$, we define *its exposure set* $E_i \subseteq \{0,1\}^M$ as the subset of diversion units such that $Y_i$ depends only on $\mathbf{Z}_{E_i} = \{Z_j : j \in E_i\}$. In other words, an outcome unit $i$'s expected outcome is entirely determined by the assignment to

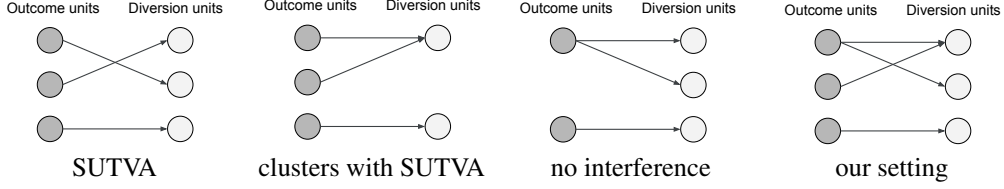

| SUTVA | clusters with SUTVA | no interference | our setting |

treatment and control of its exposure set $E_i$. In the bipartite graph representation, $E_i$ is the set of neighboring diversion units of outcome unit $i$. Similarly, we define *the influence set $I_j$* of a diversion unit $j$ as the neighboring outcome units of unit $j$ in the bipartite graph.

Recall the definition of the standard unit treatment value assumption (SUTVA) (Imbens and Rubin, 2015), also known as the individualistic treatment response (Manski, 2013), in the traditional setting:

> *The standard unit treatment value assumption (SUTVA) holds if (i) there is only one form of treatment and one form of control, (ii) a unit's outcome is unaffected by the assignment of other units to treatment or control.*

We distinguish four cases in the bipartite setting:

1. All exposure sets $E_i$ and all influence sets $I_j$ are singletons: the mapping of outcome units to diversion units is one-to-one, known as a perfect matching in the graph algorithm literature. While the diversion and outcome units remain distinct physically, we can consider each matched pair as the units of interest for the purpose of the analysis, for which SUTVA holds, and the traditional causal framework applies straightforwardly.

2. All exposure sets $E_i$ are singletons: the mapping of outcome units to diversion units is many-to-one. There is only one form of treatment and control for each outcome unit. This corresponds to Fig. 1.a of Zigler and Papadogeorgou (2018), which they call the *clusters with SUTVA* setting. When considering the disconnected components of the bipartite graph as the causal units of interest, cluster-based analysis methods (Eckles et al., 2017) apply.

3. All influence sets $I_j$ are singletons: the mapping of diversion units to outcome units is one-to-many. Because all exposure sets are disjoint from one another, we say there is *no interference* between outcome units. When assigning the diversion units to treatment and control grouped by influence sets, SUTVA holds and the traditional causal framework applies.

4. In the most general setting, the mapping of diversion units to outcome units is many-to-many, and no trivial grouping of diversion and outcome units exists.

Zigler and Papadogeorgou (2018) consider a particular setting of partial interference, where the bipartite graph can be broken up into multiple connected components. In this paper, we consider the most general setting for which no immediate reduction to the standard causal setting exists.

## 2.2 The linear treatment exposure assumption

In contrast to traditional randomized experiments where SUTVA holds and each unit only has two potential outcomes, bipartite randomized experiments are more difficult to analyze due to the large number of potential outcomes per outcome unit: $2^{|E_i|}$ for outcome unit $i \in [1, N]$. In particular, experimenters may wish to define estimands from potential outcomes that are very rarely observed. Consider, for example, the following natural extension of the average treatment effect estimand $\tau$ to the bipartite setting,

$$\tau = \sum_{i=1}^{N} Y_i(\mathbf{Z}_{E_i} = \vec{1}) - Y_i(\mathbf{Z}_{E_i} = \vec{0}) \tag{1}$$

If there exists an outcome unit $i$ for which $Pr(\mathbf{Z}_{E_i} = \vec{0}) = 0$ or $Pr(\mathbf{Z}_{E_i} = \vec{1}) = 0$, then the average treatment effect $\tau$ is not identifiable without additional assumptions on the structure of potential outcomes.

A similar problem is encountered in the non-bipartite setting when interference is present. This causal literature has focused on restrictions of the space of potential outcomes in order to make precise inference possible. For example, the *anonymous interactions* assumption (Manski, 2013) states that a unit's outcome is unchanged for any permutation of treatment assignments of its direct neighbors in an interference graph. Another popular model (Hudgens and Halloran, 2008; Toulis and Kao, 2013; Pouget-Abadie et al., 2018) assumes each unit is exposed to a direct effect (depending on its treatment assignment) and an indirect effect (depending on the proportion of treated neighboring units in an interference graph). Zigler and Papadogeorgou (2018) apply this outcome model to the bipartite setting, decomposing outcomes into the sum of a *direct* effect—the result of treating the diversion unit 'associated' with the outcome unit—and an *indirect* effect, proportional to the number treated units in its exposure set.

We consider a slight variation of their model, also studied in Toulis and Kao (2013) in the non-bipartite setting with interference, by assuming that an outcome unit's outcome is determined by a weighted proportion of the treated diversion units in its exposure set. In other words, let $w_{ij} \in R$ be the weight of the edge between diversion unit $j$ and outcome unit $i$, such that under the *linear treatment exposure model*,

$$\forall i \in [1, N], \ Y_i(\mathbf{Z}) = Y_i(\mathbf{Z}_{E_i}) = Y_i(e_i(\mathbf{Z})), \text{ where } e_i(\mathbf{Z}) = \sum_{j=1}^{M} w_{ij} Z_j$$

We call $e_i$ the *treatment exposure* of outcome unit $i$ under assignment vector $\mathbf{Z}$. The linear treatment exposure model reduces the dependence of potential outcomes on $\mathbf{Z}$ to a single scalar $e(\mathbf{Z}) \in R_+$, by assuming that each treated diversion unit contributes an additive treatment exposure to the outcome units in its influence set. Under the linear treatment exposure assumption, the analysis of bipartite experiments is very similar to the dose-response analysis commonly found in health and educational studies (Hong and Raudenbush, 2005; Moodie and Stephens, 2012; Kluve et al., 2012), which extends the causal inference literature to continuous treatment values.

### 2.3   Estimands and inference

Analogously to the dose-response literature, there are several estimands that may interest an experimenter analyzing a bipartite experiment under the linear treatment exposure assumption. Examples of such estimands are the average-exposure-response function $\mu : e \mapsto E[Y_i(e)]$ or the effect of increasing exposure by a set amount $\mu(e_0 + \partial e) - \mu(e_0)$. Some experimenters may even wish to make a *normalized exposure assumption*: $\forall i, \sum_j w_{ij} = 1$, such that $e_i(\mathbf{Z}) \in [0, 1], \forall i, \mathbf{Z}$. Under such an assumption, the extension of the commonly-used average treatment effect to the bipartite setting in Eq. 1 can be neatly rewritten

$$\tau = \frac{1}{N} \sum_i Y_i(e_i = 1) - Y_i(e_i = 0) = \mu(1) - \mu(0). \tag{2}$$

A powerful framework for inference on causal estimands is model-based imputation (Imbens and Rubin, 2015; Galagate, 2016): covariates $\mathbf{X}_i$ are collected for each outcome unit, and a parametric outcome model is assumed $Y_i(\mathbf{Z}) = f(e_i(\mathbf{Z}), \mathbf{X}_i)$, for an appropriately chosen function $f$. Once an approximating function $\hat{f}$ of $f$ is learned from the data, the estimator $\hat{\mu}(e) = 1/N \sum_i \hat{f}(e, X_i)$ can be used. For example, a simple causal-regression-based procedure to estimate the extended average treatment effect $\tau$ of Eq. 1 is:

**Step 1** Find $\hat{\alpha}, \hat{\beta}$ such that $\sum_{i=1}^{N} \left( Y_i - \alpha e_i - \beta^T X_i \right)^2$ is minimized.

**Step 2** Return $\hat{\tau} = 1/N \sum_{i=1}^{N} \left( \hat{\alpha} + \hat{\beta}^T X_i \right) - 1/N \sum_{i=1}^{N} \hat{\beta}^T X_i = \hat{\alpha}$.

This method can be sensitive to model-misspecification. In particular, treatment exposures are not identically distributed and thus realized treatment exposures may be correlated with potential outcomes. For a Bernoulli randomized design that assigns diversion units to treatment independently with probability $p$, the treatment exposure of outcome unit $i$ has expectation $E_{\mathbf{Z}}[e_i(\mathbf{Z})] = p \sum_j w_{ij}$ and $\text{Var}_{\mathbf{Z}}[e_i(\mathbf{Z})] = p(1-p) \sum_j w_{ij}^2$, such that even under a normalized exposure assumption, the distribution of treatment exposures depends on the bipartite graph.

From the dose-response literature, Imai and Van Dyk (2004) and Hirano and Imbens (2004) suggest including generalized propensity scores as covariates in the regression or stratifying on these scores for provably-consistent inference. Zigler and Papadogeorgou (2018) rely on inverse-probability-of-treatment-weighted estimators, which are also provably consistent under a partial interference assumption. These methods may not necessarily account for the correlation between two outcome units' exposure if their exposure sets are not disjoint. Further work is needed beyond the scope of this paper. See the appendix for more details.

## 3 Clustering for bipartite experiments

### 3.1 A new clustering objective

As seen in the previous section, the analysis of bipartite randomized experiments is strongly tied to the treatment exposure distribution received by outcome units. This suggests that tuning the statistical properties of this distribution might allow us to substantially improve the statistical inferences supported by the experiment. This section proposes such an optimization.

Whether the objective is to learn the average-exposure-response function or the average treatment effect estimand in Eq. 2, inferences should intuitively benefit from observing as wide a range of treatment exposures as possible across all outcome units, rather than a concentrated set of values around their expectation. One way to increase this range is to choose a design that maximizes the empirical variance of the treatment exposure vector. Cluster randomized designs, which assign treatment and control to groups of units, are a particularly popular class of designs because they do not rely too strongly on any single model-based estimator, and yet have been shown to improve variance and interference-based bias under certain conditions (Ugander et al., 2013; Gui et al., 2015; Eckles et al., 2017; Saveski et al., 2017).

We propose to choose the clustering of diversion units that maximizes the empirical variance of the treatment exposure vector $\mathbf{e}(\mathbf{Z}) = \{e_i(\mathbf{Z})\}_i$, defined as $1/N(\mathbf{e} - \bar{\mathbf{e}})^T(\mathbf{e} - \bar{\mathbf{e}})$. Since this empirical variance is a random variable, which we would like maximized for as many values of the treatment assignment vector $\mathbf{Z}$ as possible, we consider maximizing its expectation across treatment assignments as a clustering objective for diversion units:

$$\Delta = E_{\mathbf{Z}}\left[\frac{1}{N}(\mathbf{e} - \bar{\mathbf{e}})^T(\mathbf{e} - \bar{\mathbf{e}})\right] = E_{\mathbf{Z}}\left[\frac{1}{N}\sum_{i=1}^{N}(e_i(\mathbf{Z}) - \bar{e}(\mathbf{Z}))^2\right] \qquad (3)$$

An easier objective could have been to maximize the sum of individual treatment exposure variances over all outcome units. It is easy to see why this strategy fails in non-trivial bipartite graphs: assigning all diversion units to the same cluster maximizes the individual variances of each outcome unit's treatment exposure, but results in only two possible treatment exposure vector $\mathbf{e}(\mathbf{Z} = \vec{0}) = \vec{0}$ and $\mathbf{e}(\mathbf{Z} = \vec{1})$, which provides no useful basis for making causal claims.

We now provide some justifications for the empirical variance objective $\Delta$ in Eq. 3.

**Proposition 1.** *Let $M$ be the number of diversion units and $p$ the probability of assigning a diversion unit to treatment: $\Delta \leq p(M-1)/M + p^2/M$. If this upper-bound is met, then there exists an unbiased estimator of the average treatment effect.*

Thus, in the unlikely scenario that the empirical variance maximization objective achieves the upper-bound in Proposition 1, we can obtain an unbiased estimate of the average treatment effect from a resulting cluster randomized assignment. By invoking Cramer-Rao, we show that the variance maximization objective can be interpreted as an information-theoretic lower bound to certain estimators.

**Proposition 2.** *Suppose an outcome unit $i$'s response is linear in its treatment exposure $e_i$ and a set of covariates $\mathbf{X}_i \in R^d$: $Y_i(\mathbf{Z}) = \alpha e_i(\mathbf{Z}) + \beta^T\mathbf{X}_i + \epsilon_i$, where $\alpha \in R$, $\beta \in R^d$, $\epsilon \sim \mathcal{N}(0, \sigma^2)$ for $\sigma^2 \in R_+$, independent of $\mathbf{Z}$. For any unbiased estimator $\hat{\tau}$ of the average treatment effect, $Var_{\mathbf{Z},\epsilon}[\hat{\tau}] \geq \sigma^2/\Delta$ and there is equality for the ordinary least square estimator of $\alpha$ as an unbiased estimator for the average treatment effect.*

In other words, if outcomes are linear in the treatment exposure with Gaussian noise, our clustering objective minimizes a lower-bound on the variance of estimators of the average treatment effect.

## 3.2 A reduction to correlation clustering

Theorem 1 establishes the reduction of our suggested empirical variance maximization objective in Eq. 3 to a well-known problem in graph theory.

**Theorem 1.** *For diversion unit $j$, consider the vector $\vec{w}_{\cdot j} = \{w_{ij}\}$ of length $N$, where $w_{ij}$ is the weight of the edge between diversion unit $j$ and outcome unit $i$. Construct a diversion-unit-only graph, such that for each diversion unit pair $(j, k)$, its edge weight is $W_{jk} = \langle \vec{w}_{\cdot j}, \vec{w}_{\cdot k} \rangle - {}^1/_N \langle \vec{w}_{\cdot j}, \vec{1} \rangle \langle \vec{w}_{\cdot k}, \vec{1} \rangle$. Let $W_{jk}^+ = \min(0, W_{jk})$ and $W_{jk}^- = \max(0, W_{jk})$ be the positive and negative edges of the diversion-unit-only graph. For a clustering $\{\mathcal{C}\}$ of the diversion units, the variance-maximization objective can be rewritten as*

$$\Delta = \alpha + \beta \left( \sum_{\mathcal{C}} \sum_{j,k \in \mathcal{C}} W_{jk}^+ - \sum_{\mathcal{C} \neq \mathcal{C}'} \sum_{j \in \mathcal{C}', k \in \mathcal{C}} W_{jk}^- \right) \tag{4}$$

*where $\alpha = p^2 - \beta \sum_{j,k} W_{jk}^-$ and $\beta = {}^{p(1-p)}/_N$ are constants with respect to the clustering.*

We wish to maximize $\Delta$ in Eq. 4 as a function of the clustering $\{\mathcal{C}\}$ of the diversion units. Up to multiplicative and additive constants $\alpha$ and $\beta$, maximizing $\Delta$ is the maximizing-agreement formulation of the correlation clustering problem.

In contrast with other popular graph partitioning objectives (k-means, k-center, or balanced partitioning), correlation clustering does not specify the number of clusters explicitly. Rather, it seeks to maximize the difference of the number of positive edges that are within-clusters with the number of negative edges that are across clusters, without specifying a fixed number of clusters. In particular, this saves the experimenter the trouble of tuning the number of clusters explicitly, a common hyper-parameter optimization problem for cluster-based randomized experiments. Like many other constrained graph clustering problems, correlation clustering is NP-hard (Bansal et al., 2002; Charikar et al., 2003; Ailon et al., 2008) and even hard to approximate (Charikar et al., 2003; Demaine et al., 2006). Aiming to solve this clustering problem on large data sets, and inspired by previously studied heuristic algorithms for this problem (Elsner and Schudy, 2009), we propose and apply a scalable heuristic clustering algorithm for efficiently maximizing $\Delta$.

## 3.3 Algorithm

To produce a clustering of diversion units, our algorithm proceeds in two steps. We first construct the diversion-unit-only graph with edge weights $\{W_{jk}\}$ as given in Theorem 1, a step we call folding. In a second stage, we apply a scalable heuristic for our correlation clustering problem.

The diversion unit graph $\{W_{jk}\}$ from Theorem 1 is a complete graph due to the subtracted term $\langle \vec{w}_{\cdot j}, \vec{1} \rangle \langle \vec{w}_{\cdot k}, \vec{1} \rangle$. To avoid $\Theta(M^2)$ space usage, we represent this term implicitly by constructing a sparse graph with edge weights $\langle \vec{w}_{\cdot j}, \vec{w}_{\cdot k} \rangle$ and node weights $\langle \vec{w}_{\cdot j}, \vec{1} \rangle$. The implicit edge weight $W_{jk}$ equals the explicitly specified edge weight $\langle \vec{w}_{\cdot j}, \vec{w}_{\cdot k} \rangle$ minus the product of the weights of the two incident nodes $\langle \vec{w}_{\cdot j}, \vec{1} \rangle$ and $\langle \vec{w}_{\cdot k}, \vec{1} \rangle$. For large and sparse graphs, with more than 1 million units, we can use sketching to minimize the $O(M^2)$ number of necessary explicit edge computations. In the dataset we evaluate on, we used a weighted MinHash implementation (Ioffe, 2010).

Next we discuss our scalable heuristic algorithm for correlation clustering. While correlation clustering is well studied in different settings (Bansal et al., 2002; Charikar et al., 2003; Ailon et al., 2008), some provable approximation algorithms only apply to special cases of the problem (Bansal et al., 2002; Ailon et al., 2008). Other logarithmic approximation algorithms for general graphs are based on solving a linear programming relaxation (Charikar et al., 2003; Demaine et al., 2006), which is hard to scale to larger data sets. Among scalable heuristic algorithms for this problem, local search algorithms have been shown to perform well in practice (Elsner and Schudy, 2009). Inspired by these results, we apply the following local search algorithm: start from singleton clusters, and apply the following local search operations until the amount of improvement of the objective is below a threshold—hinting at the convergence of the local search procedure.

- Move one node from one cluster to another cluster if the objective function improves, and
- Merge two existing clusters if the objective function improves

In practice, we maintain three data structures: a map from each vertex to its respective cluster, a map from each cluster to the vertices in it, and a map from each cluster to the total weight of the nodes in it. Using the above data structures, for any given vertex, we can compute the best cluster to move it to in time linear in its degree. Likewise, for any given cluster, we can compute the best cluster to merge with in time linear in the number of edges incident to its vertices.

## 4   Simulation study

### 4.1   The Amazon user-item review graph

In cases where interference between certain units is present, bipartite randomized experiments can reduce bias and variance by choosing different units of analysis than the units randomly assigned to treatment. This is particularly clear in marketplace experiments, where competition from sellers often violates SUTVA. Consider, for example, the case of a randomized experiment to estimate the effect of discounting certain products in an online marketplace like Amazon on purchases. Because assigning treatment at random at the user-level may result in different users seeing different prices or incentives for the same item, randomizing on items may be preferable. Experimenters can choose to measure outcomes at the item-level as standard causal inference theory would suggest, but they can also choose to measure outcomes at the user-level as suggested by the bipartite randomized framework presented here. Doing so may help avoid complex interference mechanisms.

To evaluate our clustering algorithm, we choose a publicly-available bipartite graph dataset (McAuley et al., 2015; He and McAuley, 2016), where each edge in the dataset corresponds to a user review of a product on Amazon, totaling 83M reviews made by 121k users on 9.8M items in the graph. We discard any users having made fewer than 100 reviews from this bipartite graph, such that we can infer realistic treatment exposures that the remaining users might receive during a randomized experiment on items. The resulting unweighted graph has 7.7M edges between 2.4M items (randomization units) and 34k users (outcome units). When simulating a bipartite randomized experiment on this graph, we will use items as the randomization units we seek to cluster and users as the outcome units.

### 4.2   Clustering baselines

Clustering for causal inference has previously been studied in non-bipartite settings when interference is present. Interference is often represented by a graph in which units are linked by an edge when their potential outcomes depend on each other's treatment assignment. To optimize the bias and variance of common estimators, the solution suggested in the interference literature is to run cluster randomized designs with clusters of approximately equal size that minimize the number of edges cut in the interference graph (Eckles et al., 2017; Pouget-Abadie et al., 2018). Finding these clusters is an NP-hard problem, known as balanced partitioning, for which several heuristics have been suggested (Karypis and Kumar, 1998b; Delling et al., 2012; Stanton and Kliot, 2012; Ugander and Backstrom, 2013; Tsourakakis et al., 2014; Aydin et al., 2016). Consequently, an important baseline is to apply state-of-the-art balanced partitioning algorithms to our setting.

To produce clusters of diversion units, it is possible to run balanced partitioning on the original bipartite graph by ignoring the distinction between diversion units and outcome units, and removing all outcome units from the resulting clusters in a second stage. Balance can be enforced on both items and users, separately or together, of the bipartite graph, by setting the relevant node weights to 1 and all other node weights to 0. Another way to produce clusters of diversion units is to run balanced partitioning directly on the folded diversion unit graph we construct in the first stage of our algorithm, thus replacing the second phase of our algorithm. Each run of a balanced partitioning algorithm requires the experimenter to specify the desired number of clusters. Because our suggested correlation-clustering algorithm produced 86 clusters exactly, we run the balanced partitioning baselines for different cluster cardinalities $K \in \{50, 500, 2000, 10000\}$ and pick the best one according to the mean-squared error. For their scalability, we chose to implement and compare the balanced partitioning algorithms suggested in (Karypis and Kumar, 1998a; Aydin et al., 2016), referred to as METIS and LINE respectively.

As a set of weak baselines, we also include clusterings of cardinality $K \in \{50, 500, 2000, 10000\}$ where each diversion unit is placed in a cluster at random.

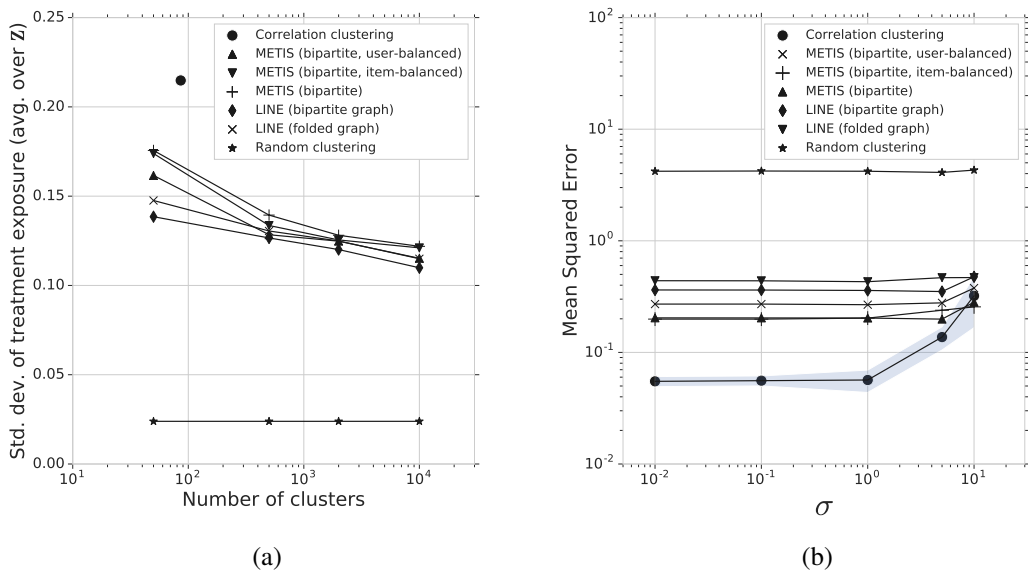

(a)                                                    (b)

Figure 1: (a) Standard deviation of the observed treatment exposure vector, averaged across assignments $\mathbf{Z}$. "bipartite" indicates the algorithm was applied to the original bipartite user-item graph; "folded" that it was applied to the folded graph from Theorem 1; "user-balanced" (resp. "item-balanced") that it enforced balance only on the user (resp. item) side of the bipartite graph. Correlation clustering does not have a cluster cardinality hyper-parameter. It produced 86 clusters. (b) Mean squared error (MSE) $\hat{\tau}$ in Step 2 of Section 2.3 for estimating the average treatment effect for varying levels of noise $\sigma$. The best performing cluster cardinality $K \in \{50, 500, 2000, 10000\}$ was chosen for each baseline in this plot. We include the $95\%$ confidence interval of our estimate of the MSE, obtained by sampling, for the correlation clustering algorithm, all other confidence intervals being of similar size.

## 4.3   Results

Because the most common A/B tests used to compare two versions of an app or an online service rarely assign treatment beyond a small subset of the general cohort, we simulate a randomized experiment on this cohort of user-items by randomly assigning $10\%$ of item-clusters to receive a simulated 'treatment'. We compute each users exposure $e_i$ to treatment as the proportion of treated items they have reviewed historically, as determined by the bipartite graph we considered in Section 4.1.

In Figure 1.a, we report the standard deviation of the observed treatment exposure vector—the square root of our optimization objective—averaged across random assignments. The correlation clustering method clearly outperforms all other baselines despite hyper-parameter tuning of the number of clusters, achieving almost $70\%$ of the highest possible value for this objective ($\sqrt{.1 \times .9} = .3$). All error bars, estimated by bootstrap, were sufficiently small to not be reported.

Furthermore, in Figure 1.b, we evaluate experimental power for each clustering—after hyperparameter optimization for all baselines—by assuming a model of potential outcomes. Sampling outcomes directly proportional to the treatment exposure received and using a causal regression analysis to determine the average treatment effect proved too easy a task. Instead, we evaluate experimental power on a misspecified model: we suppose the following linear outcome model: $Y_i \sim \sqrt{e_i} + \sigma \epsilon_i$, where $\epsilon_i \sim \mathcal{N}(0, 1)$. and use $\hat{\tau}$ given in Step 2 of Section 2.3 as an estimator of the average treatment effect $\tau$ ($= 1$) given in Eq. 1. We report the mean-squared error (MSE) of our estimator $\hat{\tau}$ defined as $E_{\mathbf{Z}, \epsilon}[(\hat{\tau} - \tau)^2]$ for different levels of noise $\sigma$. In the shaded area, we report the error bars of our correlation clustering algorithm, estimated by bootstrap. All other error bars are of equal magnitude and are not included in the figure for the sake of clarity. In low to medium noise settings ($\sigma \leq 5$), our suggested clustering achieves a significantly lower mean-squared error over other baselines.

**Acknowledgments**

The authors would like to thank Daniel Sabanés Bové and Michele Borassi for their helpful advice. We would also like to thank the anonymous reviewers for their feedback and suggestions to improve the paper.

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
