[Supplementary Material · BipartiteCausalExperimentsNeurIPSCameraReady (2).pdf]

# Supplementary materials

**Jean Pouget-Abadie**
Google Research
New York, NY 10011
jeanpa@google.com

**Kevin Aydin**
Google Research
Mountain View, CA 94043
kaydin@google.com

**Warren Schudy**
Google Research
New York, NY 10011
wschudy@google.com

**Kay Brodersen**
Google
Zürich, Switzerland
kbrodersen@google.com

**Vahab Mirrokni**
Google Research
New York, NY 10011
mirrokni@google.com

## 1   Debiasing inference with a generalized propensity score approach

When realized treatment exposures are correlated with potential outcomes, the dose-response literature has suggested generalizations of the propensity score to de-bias inference. Hirano and Imbens (2004) define the *generalized propensity score* function $r : (e, \mathbf{X}) \mapsto r(e, \mathbf{X})$ as the density of the conditional distribution of outcome unit $i$'s treatment exposure given its covariates $\mathbf{X}_i$. They suggest learning the generalized propensity score function $r(\cdot, \cdot)$ as well as the conditional outcome distribution $\beta : (e', r') \mapsto E[Y_i | e_i = e', r(e', \mathbf{X}_i) = r']$, conditioned on treatment exposure $e'$ and generalized propensity score $r'$. Finally, they propose $\hat{\mu}(e) = 1/N \sum_i \beta(e, r(e, \mathbf{X}_i))$ as an estimator for the average dose-response function $\mu$. In our case, the treatment exposure distribution of outcome unit $i$ is known and fully parameterized by its outgoing-edge weights $\{w_{ij}\}_j$.

In practice, Hirano and Imbens (2004) suggest fitting a linear regression of $Y_i$ on the realized treatment exposure and corresponding propensity score couplet $(e_i, r(e_i, \mathbf{X}_i))$ to construct an approximation $\hat{\beta}$ of the conditional outcome distribution $\beta$, necessary for computing $\hat{\mu}$. Imai and Van Dyk (2004) propose a similar approach, which stratifies outcome units into $S$ strata by any uni- or multivariate parameter $\theta_i$ such that $r(\cdot, \mathbf{X}_i) = r(\cdot, \theta_i)$, and learns $\hat{f}_s$ within each strata such that $Y_i(\mathbf{Z}) = f_s(e_i(\mathbf{Z}))$. They suggest using $\hat{f}(\cdot) = 1/S \sum_s \sum_{i \in s} \hat{f}_s(\cdot) W_s$ as an estimator for the average dose response function $\mu$, where $W_s$ is the number of outcome units in strata $S$.

## 2   Proof of Proposition 1

To prove Proposition 1, we consider a rewriting of the objective from Theorem 1.

$$\Delta = p - \frac{p(1-p)}{N} \sum_{\mathcal{C} \neq \mathcal{C}'} \sum_{j \in \mathcal{C}, k \in \mathcal{C}'} \langle \phi_j, \phi_k \rangle - \frac{p(1-p)}{N^2} \sum_{\mathcal{C}} \left( \sum_{j \in \mathcal{C}} \langle \phi_j, \vec{1} \rangle \right)^2$$

We decompose this objective term-by-term. For all diversion unit pairs $j, k \in [1, M]$, $\langle \phi_j, \phi_k \rangle \geq 0$, with equality if and only if diversion unit $j$ and diversion unit $k$ have no common outcome unit neighbors. As a result, $\sum_{\mathcal{C} \neq \mathcal{C}'} \sum_{j \in \mathcal{C}, k \in \mathcal{C}'} \langle \phi_j, \phi_k \rangle \geq 0$, with equality if and only if clusters $\mathcal{C}$ and $\mathcal{C}'$ have no common outcome unit neighbors. Furthermore, the following equality holds: $\sum_{\mathcal{C}} \sum_{j \in \mathcal{C}} \langle \phi_j, \vec{1} \rangle = N$. From the Cauchy-Schwarz inequality, $\sum_{\mathcal{C}} (\sum_{j \in \mathcal{C}} \langle \phi_j, \vec{1} \rangle)^2 \geq N^2/K$, where $K$ is the total number of clusters $\mathcal{C}$, with equality if and only if $\forall \mathcal{C}, \mathcal{C}', \sum_{j \in \mathcal{C}} \langle \phi_j, \vec{1} \rangle =$

$\sum_{j \in \mathcal{C}'} \langle \phi_j, \vec{1} \rangle = {}^N/_K$. If there exists a clustering $\{\mathcal{C}\}_K$ with $K$ clusters, such that the variance maximization objective for $\{\mathcal{C}\}_K$ is equal to $p - \frac{p(1-p)}{K}$, then $\{\mathcal{C}\}_K$ cuts no edges of the bipartite graph $G$. As a result, each outcome unit receives either treatment exposure 1 or 0 for every assignment $\mathbf{Z}$, and the stable unit treatment value assumption holds.

# 3  Proof of proposition 2

The first claim is an application of Gauss-Markov; the second claim is an application of Cramer-Rao.

For the sake of exposition, we assume that $\beta = 0$ and $\mathbf{Y} = \alpha \mathbf{e} + \epsilon$, for all assignments $\mathbf{Z}$. The average treatment effect is equal to $\alpha$. Hence, we can restate our proposition with $\hat{\tau}$ as estimators of $\alpha$.

For a fixed assignment vector $\mathbf{Z}$, the Cramer-Rao bound states that the variance of any unbiased estimator $\hat{\tau}$ of $\alpha$ is such that $\mathrm{Var}[\hat{\alpha}] \geq I(\alpha)^{-1}$, where $I(\alpha) = -E\left[\frac{\delta^2 l(\mathbf{Y}, \mathbf{e}; \alpha)}{\delta \alpha^2}\right]$ is the Fisher information of $\alpha$ and $l$ is the log-likelihood of observing $(\mathbf{Y}, \mathbf{e})$ given $\alpha$ and $\mathbf{Z}$. With $l(\mathbf{Y}, \mathbf{e}; \alpha) = -\frac{N}{2}\log(2\pi\sigma^2) + \sum_{i=1}^{N} \frac{(y_i - \alpha e_i)^2}{2\sigma^2}$, we obtain $I(\alpha) = \frac{1}{N}(\mathbf{e} - \bar{\mathbf{e}})^T(\mathbf{e} - \bar{\mathbf{e}})$. By the law of total variance:

$$\mathrm{Var}_{\mathbf{Z},\epsilon} = E_{\mathbf{Z}}[\mathrm{Var}_\epsilon[\hat{\tau}|\mathbf{Z}]] + \mathrm{Var}_{\mathbf{Z}}[E_\epsilon[\hat{\tau}|\mathbf{Z}]] = E_{\mathbf{Z}}[\mathrm{Var}_\epsilon[\hat{\tau}|\mathbf{Z}]] + \mathrm{Var}_{\mathbf{Z}}[\alpha] = E_{\mathbf{Z}}[\mathrm{Var}_\epsilon[\hat{\tau}|\mathbf{Z}]]$$

Hence the result becomes:

$$\mathrm{Var}_{\mathbf{Z},\epsilon}[\hat{\tau}] \geq E_{\mathbf{Z}}\left[\frac{\sigma^2}{\frac{1}{N}(\mathbf{e} - \bar{\mathbf{e}})^T(\mathbf{e} - \bar{\mathbf{e}})}\right] \geq \frac{\sigma^2}{E_{\mathbf{Z}}[\frac{1}{N}(\mathbf{e} - \bar{\mathbf{e}})^T(\mathbf{e} - \bar{\mathbf{e}})]}$$

# 4  Proof of Theorem 1

Let $\Phi \in R^{N \times M}$ be the adjacency matrix of the bipartite graph between diversion units and outcome units, such that $\Phi_{ij} = w_{ij}$ and $\phi_j = \vec{w}_{.j}$. Because $\mathbf{e}(\mathbf{Z}) = \Phi\mathbf{Z}$, the variance-maximization objective in Eq. 3 can be rewritten as

$$\frac{1}{N}\left(\mathbf{e}(\mathbf{Z}) - \bar{\mathbf{e}}(\mathbf{Z})\right)^T\left(\mathbf{e}(\mathbf{Z}) - \bar{\mathbf{e}}(\mathbf{Z})\right) = \frac{1}{N}\mathbf{Z}^T\Phi^T\Phi\mathbf{Z} - \left(\frac{1}{N}\mathbf{1}^T\Phi\mathbf{Z}\right)^2$$

Let $p$ be the probability that a diversion unit is assigned to treatment and $\Sigma = E_{\mathbf{Z}}[\mathbf{Z}^T\mathbf{Z}]$ be the variance-covariance matrix of $\mathbf{Z}$. Taking the expectation of the quadratic form in $\mathbf{Z}$,

$$E_{\mathbf{Z}}\left[\frac{1}{N}\mathbf{Z}^T\Phi^T\Phi\mathbf{Z}\right] = \frac{1}{N}\left(\mathrm{Tr}\left[\Phi^T\Phi\Sigma\right] + p^2\mathbf{1}^T\Phi^T\Phi\mathbf{1}\right) = \frac{1}{N}\mathrm{Tr}\left[\Phi^T\Phi\Sigma\right] + p^2,$$

where the second equality is obtained by observing that $\Phi\vec{1} = \vec{1}$. If two diversion units $j$ and $k$ belong to the same cluster $\mathcal{C}$, then $\Sigma_{jk} = p$; otherwise, $\Sigma_{jk} = p^2$. Hence,

$$\mathrm{Tr}\left[\Phi^T\Phi\Sigma\right] = \sum_{\mathcal{C}}\sum_{j,k \in \mathcal{C}^2} p(\Phi^T\Phi)_{jk} + \sum_{\mathcal{C} \neq \mathcal{C}'}\sum_{j \in \mathcal{C}, k \in \mathcal{C}'} p^2(\Phi^T\Phi)_{jk}$$

Because $(\Phi^T\Phi)_{jk} = \langle \phi_j, \phi_k \rangle$ and $\sum_{jk}\langle \phi_j, \phi_k \rangle = N$, the above becomes

$$E_{\mathbf{Z}}\left[\frac{1}{N}\mathbf{Z}^T\Phi^T\Phi\mathbf{Z}\right] = p^2 + p - \frac{p(1-p)}{N}\sum_{\mathcal{C} \neq \mathcal{C}'}\sum_{j \in \mathcal{C}, k \in \mathcal{C}'}\langle \phi_j, \phi_k \rangle$$

Taking the expectation of the second term of the objective,

$$
\begin{aligned}
E_{\mathbf{Z}}\left[\left(\frac{1}{N}\mathbf{1}^T\Phi\mathbf{Z}\right)^2\right] &= \frac{1}{N^2}\sum_{i,j,k,l}\Phi_{ij}\Phi_{lk}E_{\mathbf{Z}}[Z_jZ_k] \\
&= \frac{1}{N^2}\sum_{j,k}\langle\phi_j,\vec{1}\rangle\langle\phi_k,\vec{1}\rangle E_{\mathbf{Z}}[Z_jZ_k] \\
&= \frac{1}{N^2}\left(p\sum_{\mathcal{C}}\sum_{j,k\in\mathcal{C}}\langle\phi_j,\vec{1}\rangle\langle\phi_k,\vec{1}\rangle + p^2\sum_{\mathcal{C}\neq\mathcal{C}'}\sum_{j\in\mathcal{C},k\in\mathcal{C}'}\langle\phi_j,\vec{1}\rangle\langle\phi_k,\vec{1}\rangle\right)
\end{aligned}
$$

Noting that $\sum_{j,k=1}^{M}\langle\phi_j,\vec{1}\rangle\langle\phi_k,\vec{1}\rangle = N^2$, the previous term becomes

$$
E_{\mathbf{Z}}\left[\left(\frac{1}{N}\mathbf{1}^T\Phi\mathbf{Z}\right)^2\right] = \frac{p^2N^2}{N^2} + \frac{p(1-p)}{N^2}\sum_{\mathcal{C}}\sum_{j,k\in\mathcal{C}}\langle\phi_j,\vec{1}\rangle\langle\phi_k,\vec{1}\rangle
$$

The final objective can be written as:

$$
\begin{aligned}
\Delta &= p - \frac{p(1-p)}{N}\sum_{\mathcal{C}\neq\mathcal{C}'}\sum_{j\in\mathcal{C},k\in\mathcal{C}'}\langle\phi_j,\phi_k\rangle - \frac{p(1-p)}{N^2}\sum_{\mathcal{C}}\sum_{j,k\in\mathcal{C}}\langle\phi_j,\vec{1}\rangle\langle\phi_k,\vec{1}\rangle \\
&= p^2 + \frac{p(1-p)}{N}\sum_{\mathcal{C}}\sum_{j,k\in\mathcal{C}}\left(\langle\phi_j,\phi_k\rangle - \frac{1}{N}\langle\phi_j,\vec{1}\rangle\langle\phi_k,\vec{1}\rangle\right)
\end{aligned}
$$

Let $W_{jk} = \langle\phi_j,\phi_k\rangle - \frac{1}{N}\langle\phi_j,\vec{1}\rangle\langle\phi_k,\vec{1}\rangle$, $W_{jk}^{+} = max(0,W_{jk})$ and $W_{jk}^{-} = min(0,W_{jk})$ be the positive and negative edges of the graph respectively, and $W^{-} = \sum_{j,k}^{N}W_{jk}^{-}$ be the sum of all negative edges in the graph. The objective becomes:

$$
\Delta = p^2 + \frac{p(1-p)}{N}\left(W^{-} + \sum_{\mathcal{C}}\sum_{j,k\in\mathcal{C}}W_{jk}^{+} - \sum_{\mathcal{C}\neq\mathcal{C}'}\sum_{j\in\mathcal{C}',k\in\mathcal{C}}W_{jk}^{-}\right)
$$