[Reviews · NeurIPS 2019]

Reviewer 1



This work addresses the task of designing experiments on bipartite graphs in the presence of SUTVA violation. Specifically, the authors propose an extension to the graph cluster randomization framework to the bipartite setting by casting the problem as correlation clustering. Propositions are provided that tie the variance of the causal estimate to the quality of the discovered solution. A heuristic is provided to solve the correlation clustering problem. Experiments are provided comparing the proposed method to balanced partitioning and random assignment which show the proposed method performing favorably. Overall, I think this is a nice, simple, solution to a problem that occurs a fair amount for practitioners. The paper is clearly written and motivated. The connection to correlation clustering is sensible and provides a decent interpretation of the results. I have two is that there are details missing from the experiment section that would improve reproducibility. Specifically the authors do not state how the estimation is carried out. It would be nice to see comparisons to both the estimator of Eckles, et al. (2017) and Gui, et al. (2015). A few comments / questions: 1. In the introduction the connection to optimal experimental design is alluded to. Having a more precise connection made within the paper would be both interesting and helpful. 2. It is not clear what the quality of the approximation is for the heuristic presented. It would be nice to see a discussion of this and the implications for the variance of the estimator. 3. What is the space complexity of the proposed heuristic? 4. Why are other correlation clustering heuristics not compared against in the experiment section? 5. There are no results given for the variance of the estimators / designs. Given that this is often a critical component of analysis for practitioners it would be great to estimates (and ideally coverage) for each of the shown methods, as well as a comparison that shows empirically the relationship between approximation quality of the correlation clustering heuristic and the resulting variance.

Reviewer 2



This work addresses an important and interesting setting, bipartite experiments, for causal inference in randomized experiments. In such a setting, treatment and control units are different from the outcome units. Unlike the prior work from Zigler and Papadogeorgou (2018) that considers a particular setting of partial interference, they consider the most general setting for which no immediate reduction to the standard causal setting exists. Moreover, they assume that an outcome unit’s outcome is determined by a weighted proportion of the treated diversion units in its exposure set. The main novelty is that they propose a new clustering objective, i.e., choose the clustering of diversion units that maximizes the empirical variance of the treatment exposure. To solve the optimization problem, they propose a scalable heuristic clustering algorithm. They validate the approach by running experiments on real data. The paper is well written and easy to follow. The reviewer thinks this work is interesting and insightful for future work in this direction.

Reviewer 3



The paper discusses the analysis of bipartite randomized experiments, where treatment is assigned to diversion units and outcomes are measured on outcome units. The ideas discussed propose the use of correlation clustering to identify clusters of diversion units to benefit the estimation of a treatment effect. A scalable algorithm is provided for the correlation clustering and there is a simulation example to compare the variance of estimates when using other approaches to clustering. The paper clearly demonstrates the benefits and popularity of bipartite experiments and explains the need for clustering. The level of novelty may be questioned as it is unclear of the benefits of the proposed algorithm versus other algorithms. There is some novelty to the ideas presented and the algorithm developed would be useful in various applications, but a more thorough explanation of novelty may be required. There is no conclusion section.

[Author Response · NeurIPS 2019]

We would like to thank all three reviewers for their careful review and constructive feedback. We have addressed points made by reviewers 1 and 3 below. We added a reference to (Basse and Airoldi, 2015); other references can be found in the main paper.

## Reviewer 1

*" In the introduction the connection to optimal experimental design is alluded to."* Thank you for this feedback. If accepted, we would be happy to detail the connection with the optimal experimental design literature. Because our cluster randomized design is chosen so as to optimize a statistical criterion (the empirical exposure variance) within the class of cluster randomized designs, our approach fits squarely within the optimal experimental design literature. Canonical work by Raudenbush (1997) explores optimal cluster randomized designs under a different optimization objective than ours since no violations of SUTVA are considered in their paper. More recently, Basse and Airoldi (2015) consider a non-bipartite setting where interference is present but explores optimal allocation strategies within a given graph, rather than finding an appropriate clustering. Like many traditional work on optimal designs, our objective is tied to the Fisher information matrix of an inference problem. In fact, conditionally on a random assignment and under a linear exposure model with Gaussian noise, maximizing the Fisher information matrix is equivalent to maximizing the empirical exposure variance, as detailed in the proof of Proposition 2.

*"It is not clear what the quality of the approximation is for the heuristic presented [...]"* We agree that the quality of the approximation for the proposed heuristic could have been made clearer. As a measure of quality, we can compute how well our heuristic approximates an upper-bound of the objective. When treating $10\%$ of all diversion units, the maximal exposure variance we can hope for is that each outcome unit individually gets exposure 1 with probability .1 and 0 otherwise. The value of this optimistic upper-bound is $\sqrt{.1 \times .9} = .3$. The objective value of .23 that we achieved with our algorithm and reported in Fig. 1.a. is therefore a $76\%$ approximation of the upper-bound.

*"What is the space complexity of the proposed heuristic?"* Thank you for bringing this omission to our attention, which we would be happy to clarify in a final version of the paper if accepted. The folding stage has worst-case $O(M^2)$ space complexity if the resulting folded graph is complete, where M is the number of diversion units. In practice, we found that most folded graph were sparse and that filtering low-weight edges in certain graphs did not substantially affect the quality of the final solution, thus reducing the practical space complexity. The correlation clustering heuristic stage uses space linear in the size of the folded graph since each of the maintained data structures has size linear in the number of diversion units (cf. l. 264), plus the adjacency list of the folded graph.

*"Why are other correlation clustering heuristics not compared against in the experiment section?"* During the development phase, we explored several variations of the local search heuristic before settling on the one presented in the paper. We chose local search because it has been shown to perform well in practice, while being very scalable unlike semi-definite-programming-based solutions (Elsner and Schudy, 2009). If accepted, we would be happy to include comparisons to other attempted local search heuristics.

*"There are no results given for the variance of the estimators / designs [...]"* Thank you for this feedback. By reporting the mean-squared error of the estimators / designs in Figure 1.b., we include both bias and variance under one metric: MSE = bias$^2$ + variance. If accepted, we would be happy to plot bias, variance, and coverage in separate figures. We would also be happy to plot the empirical relation between each of the three objective values for each algorithm: the maximizing-agreement correlation-clustering objective, the empirical exposure variance, and final estimator variance.

## Reviewer 3

*"[. . . ] a more thorough explanation of novelty may be required."* We have evidence of significant improvements using this methodology over previous baselines in a real-world setting. Unfortunately, that dataset cannot be made public. We repeated the experiment on a publicly-available Amazon dataset for which we saw clear gains over prior art.

*"There is no conclusion section."* Due to space constraints, we did not include a conclusion section but we would be happy to include one that recaps the main contributions of the paper and suggests future work: understanding how estimation can be improved with generalized propensity scores and finding a similar algorithm to cluster the outcome units are two directions that come to mind.

## References

Guillaume W Basse and Edoardo M Airoldi. 2015. Optimal design of experiments in the presence of network-correlated outcomes. *ArXiv e-prints* (2015).


[Meta-Review · NeurIPS 2019]

As pointed out by the reviewers, these are the strengths and weaknesses of the paper: STRENGTHS The paper proposes a method for bipartite experiment design building upon previous work on graph cluster randomization. It presents a novel optimization function and a heuristic to solve it. The method is shown to lead to balanced partitioning using empirical evaluation. The paper is well-written and easy to follow, and the reviewers had a positive discussion about it. FOR IMPROVEMENT The main concerns that need to be addressed to strengthen this paper include insufficient experimental comparisons (R1,R3), unclear connection made to optimal experimental design (R1), unclear relationship between clustering quality and estimate variance (R1).